# Strong tidal variations in ice flow observed across the entire Ronne Ice Shelf and adjoining ice streams

Sebastian, H. R. Rosier<sup>1</sup>, G. Hilmar Gudmundsson<sup>1</sup>, Matt, A. King<sup>2</sup>, Keith, W. Nicholls<sup>1</sup>, Keith Makinson<sup>1</sup>, and Hugh, F. J. Corr<sup>1</sup>

<sup>1</sup>British Antarctic Survey, High Cross, Madingley road, Cambridge, UK

<sup>2</sup>University of Tasmania, Hobart, Tasmania, Australia

Correspondence to: S. H. R. Rosier (s.rosier@bas.ac.uk)

Abstract. We present a compilation of GPS time series, including those for previously unpublished sites, showing that flow across the entire Ronne Ice Shelf and its adjoining ice streams is strongly affected by ocean tides. Previous observations have shown strong diurnal and semidiurnal motion of the ice shelf and surface flow speeds of Rutford Ice Stream (RIS) are known to vary with a fortnightly ( $M_{\rm sf}$ ) periodicity. Our new dataset shows that the  $M_{\rm sf}$  flow modulation, first observed on RIS, is also

- 5 found on Evans, Talutis, Institute and Foundation Ice Streams, i.e. on all ice streams for which data are available. The amplitude of the  $M_{\rm sf}$  signal increases downstream of grounding lines, reaching up to 20% of mean flow speeds where ice streams feed into the main shelf. Upstream of ice stream grounding lines, decay length scales are relatively uniform for all ice streams but the speed at which the  $M_{\rm sf}$  signal propagates upstream shows more variation. Observations and modelling of tidal variations in ice flow can help constrain crucial parameters that determine the rate and extent of potential ice mass loss from Antarctica.
- 10 Given that the  $M_{\rm sf}$  modulation in ice flow is readily observed across the entire region, at distances of up to 80 km upstream of grounding lines, but is not completely reproduced in any existing numerical model, this new dataset suggests a pressing need to identify the missing processes responsible for its generation and propagation. The new GPS data set is publicly available through the UK Polar Data Centre at http://doi.org/10.5285/4fe11286-0e53-4a03-854c-a79a44d1e356.

# 1 Introduction

15 Ocean tides affect the flow of ice shelves (Doake et al., 2002; Brunt et al., 2010; Makinson et al., 2012; Legresy et al., 2004; King et al., 2011a) and ice streams (Bindschadler et al., 2003b, a; Anandakrishnan et al., 2003; Anandakrishnan and Alley, 1997; Gudmundsson, 2006; Marsh et al., 2013), often far upstream of their grounding lines (GLs). The growing availability of high temporal resolution data, particularly using GPS, has made the extent of these tidal effects increasingly apparent.

The Weddell Sea region is an ideal place to look for tidal effects on ice flow as it is subjected to the largest tides in Antarctica, with a tidal range of up to 9 m at the GL of ice streams along the Zumberge coast. The semidiurnal  $M_2$  (principal lunar) and  $S_2$  (principal solar) tides dominate the vertical tidal signal beneath most of the ice shelf and are characterised by an amphidromic point at the Ronne Ice Shelf calving front (Fig. 1a, Padman et al. 2002). An amphidromic point (or tidal node) is a point at which a tidal constituent has zero amplitude, around which that constituent rotates. The diurnal tides are smaller

**Figure 1.** Amplitudes of the semidiurnal  $M_2$  (panel a) and diurnal  $K_1$  (panel b) tides in the Weddell Sea region. Dashed lines are 20° phase lines (also labelled in degrees), with the  $M_2$  amphidromic point visible as the minimum in amplitude in panel a. Note the difference in colour scale between the two panels. Amplitudes and phases are taken from the Circum-Antarcitc inverse Model (CATS2008a, Padman et al. 2002)