# Peer review of "Strong tidal variations in ice flow observed across the entire Ronne Ice Shelf and adjoining ice streams"

_Earth System Science Data, 2017_

## Referee Comment (RC1) · L. Padman (Referee) · 26 Jul 2017

Review of: Strong tidal variations in ice flow observed across the entire Ronne Ice Shelf and adjoining ice streams, by S. Rosier and others.

SUMMARY

This paper describes an extensive GPS data set of 3-D ice shelf motion for the Ronne Ice Shelf. It focuses on introducing the data set (43 total GPS sites on floating and grounded ice sheet) and describing the lateral motion with an emphasis on the Msf tidal component. Interpretation is limited to reviewing previous studies relating the low-

frequency tidal signals to the standard vertical tidal components, but this is appropriate to ESSDD.

Specific comments are divided into "Major", where the authors should explain what they did in response, and "Minor", which don't require documentation in revision. Numbers refer to original page.line. In general the paper is very clear and well written.

– Laurie Padman

GENERAL COMMENTS

1. I'd prefer that "shelf" always be "ice shelf"; however, in this paper it isn't critical.

2. Say '. . . BY Makinson et al. . . .' rather than '. . . IN Makinson et al. . . .' ?

MAJOR COMMENTS

MSF is the interaction of principal semidiurnals M2 and S2. This needs to be explained, since the diurnals K1 and O1, while smaller (in general) in the Weddell Sea, are still significant, and their interaction gives MF. Maybe the records are generally too short to deconvolve Msf and Mf, but it seems like Mf should be there. I also seem to recall papers about diurnals being stronger in lateral motion on grounded ice than semidiurnals, relative to the nearby ocean vertical signal, which suggests that maybe Mf would be amplified relative to Msf, at least on grounded ice?

General: I think you need to be clear that Talutis and Carlson ice streams flow into what you only call "Carlson Inlet".

5.1-5.2: This seems like a good place to explain, or reiterate, that Msf does have the same frequency as the M2/S2-based spring-neap cycle, since it hints at why the nonlinearity that appears as Msf lateral motion might arise.

7.11-7.23: (a) This would be clearer if it was organized in frequency: M2, K1, Msf, Ssa. Maybe a short paragraph for each. (b) Then, organize Fig. 5 to the write-up, and cite the specific panel we're meant to look at for each.

7.26: "As the tide rotates". Which is "the tide" here? Msf, or general energy flux of the principal tides, mainly M2 and S2? But maybe you mean "Although" rather than "As" ? I'm trying to understand how the tidal rotation relates to the lead/lag between the spring-neap modulation and Msf phase, without invoking physics that you haven't explained.

11.1-11.6: I'd disagree with the idea that interpolated fields "better show the importance of". Figure 5a is the honest answer. Maybe if you just used a wider color range for Fig. 5 ("jet" rather than new Matlab default), you could easily discuss everything from Fig. 5a.

11.7-11.9: As for the previous comment, the dot plots of Fig. 5 are more honest than the interpolated map, so I'd rather see the dot plot for MSF velocity (as a fraction of mean speed).

11.7-11.16: I didn't understand this entire paragraph. (a) If the record is long enough to resolve Msf, then the detided signal has Msf removed, so the mean flow *based on the detided residual* should be unbiased by non-integer numbers of Msf in the record interval. The problem only arises if you do a standard mean of the non-detided record. (b) Why would the problem be worse in slower-moving parts of the ice sheet, especially if there is a reasonably linear relationship between Msf amplitude and mean speed (Figure 9)? (c) I really don't find Fig. 8b convincing, as the structure seems to be dominated by regions with poor data constraints.

11.17-12.3: I think we need a bit more information here about *why* fortnightly variation *always* leads to increased mean speed regardless of mechanism. The three cited papers are all about ice *streams*; how does that map to, say, the Ronne ice front? Do you think that the Msf strain rates on the floating ice are increasing the mean flow, and increasing the mean strain?

12.10-12.12: I wouldn't say that "the Ssa tides *leads to* a maximum in S2: they are part of the same component of astronomical forcing, and the description only makes

sense if you are thinking S2 as being some modulated addition of S2 and K2.

12.19-12.20: My guess is that the statement that "FRIS is ... the best observed ... system" only really applies to the size of the GPS data base. For different aspects of ice shelves, more complete knowledge exists. It is probably good enough to just say that "The extensive data set of GPS records, and the large tidal ranges, ... make the FRIS an excellent place to ..."

12.28-12.29: "has been shown to lead to a higher ..." Not true *of this paper*, and maybe not true in general for the ice shelf if the only basis is glaciological models for ice streams?

MINOR COMMENTS

1.3: (a) "motion" (of the ice shelf) in what direction? Clearly in the vertical, but less in the horizontal. (b) comma after 'shelf'

1.10-1.13: Reorganize and/or subdivide the sentence.

1.20: cite for the 'up to 9 m' statement. I think this comes from interpreted tilt rather than GPS.

4.15-4.18: This paragraph should be merged with lines 4.3-4.7, to put the observations stuff together before summarizing the hypotheses for nonlinear effects.

4.28: "overwinter" seems like an adjective. So, "continuous overwinter data ..." or "data collection over the winter"

4.33: Don't need "It is important to emphasise that"

5.7 and 5.11: Why 'FR-' sites?

5.12: Less colloquial way to say "these steps were skipped"

6.1: When you say that the Rayleigh criterion (lower case c) "was used", do you mean that it's a characteristic of the software, or *you* made the decision to use it?

[Figure]

7.30: No need to hyphenate "time scales"

8.2-8.3: Sentence starting "The various" is just figure caption material.

8.4: capitalization of "data analysis" ?

10.6: Bracketing of citation "(Minchew et al., 2016)"

11.9: formatting of 'Msf'

12.16-12.17: This interesting issue (aliasing of InSAR-derived velocities) seems like it should have been mentioned in the Introduction.

FIGURES

F.1: (a) Add ice front to plot. Correct citation for CATS2008a is "an updated version of the inverse tide model described by Padman et al. (2002)"

F.2: (a) Caption should explain ice stream names, including that box-d is Talutis and Carlson ice streams. (b) "smoothed with a low-pass boxcar filter" should probably explain its characteristics (X km) and *brief* reason why.

F.3: Part of the 'TIS' box is actually also "Carlson Ice Stream", isn't it?

F.6: For reasons I can't explain, I expected grounded ice to be on the *left* side of these plots. You could reverse them for me. Or, probably more sensible, add a vertical dashed line on each panel for the GL, then at the top of the plot mark "Ice Stream" and "Ice Shelf".

F.8: See Major Comments. I don't think this interpolated plot works well; the dots like Fig. 5 are better, especially for panel (b)

F.9: This might be more information-rich if you used different symbols for ice shelf and ice streams.

---

## Referee Comment (RC2) · R. Walker (Referee) · 5 Sep 2017

This is an intriguing data set that I look forward to checking out for myself.

I have several comments, all of them relatively minor issues.

Page 1, Line 15) The citations are out of order here (and a few other places).

Figure 1) I'd like to see the ice front marked on this, if it doesn't make the figure harder to read. If that doesn't work, please give the distance between the amphidrome and ice front.

Figure 2) At least on the version I printed out, there's not that much color contrast

[Figure]

between medium and dark blue. Also, it would be good to give the ice stream names in this caption, so it's not necessary to look ahead to Figure 3.

Figure 3) Could you label at least one contour so it's easier to see ice velocity?

Page 3, Line 4 ff.) This paragraph should also mention Anandakrishnan et al (2003) on Bindschadler ice stream, which is a case of more or less diurnal forcing causing more or less diurnal response. Also, it's a bit odd to refer to FRIS in a paragraph about behavior different from your observations here, considering that FRIS is part of your domain.

Figure 7) Which components had SNR > 2? Just the ones in Table 1?

Page 12, Line 5) Carlson Inlet isn't shown on any of the maps.

Page 12, Line 6) More quantitatively, what are you considering to be fast or slow flow?

---

## Author Comment (AC1) · 28 Sep 2017

We thank Ryan Walker for his helpful comments and suggestions for our manuscript. We have included his points below in italics, together with our response to each point in bold.

*Page 1, Line 15) The citations are out of order here (and a few other places).*

**Citations are now ordered consistently throughout**

*Figure 1) I'd like to see the ice front marked on this, if it doesn't make the figure harder to read. If that doesn't work, please give the distance between the amphidrome and*

*ice front.*

**Added the ice front as a magenta line**

*Figure 2) At least on the version I printed out, there's not that much color contrast between medium and dark blue. Also, it would be good to give the ice stream names in this caption, so it's not necessary to look ahead to Figure 3.*

**Done**

*Figure 3) Could you label at least one contour so it's easier to see ice velocity?*

**Added labels to the ice velocity contours**

*Page 3, Line 4 ff.) This paragraph should also mention Anandakrishnan et al (2003) on Bindschadler ice stream, which is a case of more or less diurnal forcing causing more or less diurnal response. Also, it's a bit odd to refer to FRIS in a paragraph about behavior different from your observations here, considering that FRIS is part of your domain.*

**A sentence has been added, discussing the Bindschadler ice stream observations. With regards to the second comment, this paragraph is not aimed at highlighting different responses from our own observations but is giving an overview of the tidal response of ice streams/shelves around Antarctica. We dedicate a separate paragraph talking about the very different Msf response because that is largely our focus but we do also go on to discuss other tidal frequencies such as the semidiurnals and diurnals.**

*Figure 7) Which components had SNR > 2? Just the ones in Table 1?*

**What we mean here is that the reconstructed displacements are made by including every tidal constituent with an SNR > 2 as determined from the original data for each site. Re-worded slightly to make this clearer.**

*Page 12, Line 5) Carlson Inlet isn't shown on any of the maps.*

**This is now named in Fig 3b.**

*Page 12, Line 6) More quantitatively, what are you considering to be fast or slow flow?*

**Given that this is a very general statement we don't feel that attributing specific cut-off values is appropriate. Presumably with a much higher measurement precision the Msf signal would be observable even on slower moving portions of the shelf and coastal ice streams. Changed the wording slightly to hopefully make our point clearer.**

―――――――――――――――――

---

## Author Comment (AC2) · 28 Sep 2017

We thank Laurie Padman for his thorough and thoughtful comments on our manuscript. We have included all of his points below in italics, together with our response to each point in bold.

GENERAL COMMENTS

*1. I'd prefer that "shelf" always be "ice shelf"; however, in this paper it isn't critical.*

**Replaced all instances of 'shelf' with 'ice shelf'**

*2. Say '. . . BY Makinson et al. . . .' rather than '. . . IN Makinson et al. . . .' ?*

[Figure]

Changed all instances of 'by . . . et al.' to 'in . . . et al.'

MAJOR COMMENTS

*MSF is the interaction of principal semidiurnals M2 and S2. This needs to be explained, since the diurnals K1 and O1, while smaller (in general) in the Weddell Sea, are still significant, and their interaction gives MF. Maybe the records are generally too short to deconvolve Msf and Mf, but it seems like Mf should be there. I also seem to recall papers about diurnals being stronger in lateral motion on grounded ice than semidiurnals, relative to the nearby ocean vertical signal, which suggests that maybe Mf would be amplified relative to Msf, at least on grounded ice?*

**We have added the following paragraph in the discussion: In the same way that an interaction between the two principal semidiurnal constituents (M2 and S2) leads to an Msf signal in ice flow, the two principal diurnal constituents (K1 and O1) will interact nonlinearly to produce an Mf response. This is somewhat difficult to separate from the Msf signal because they have a similar periodicity (Table 1). Given that the semidiurnal constituents are generally much larger than the diurnals beneath the FRIS we can be confident that the Msf response is most important, but there is likely to be some response in ice flow at the Mf frequency (King et al., 2010). Indeed, at the GL of the Ross Ice Shelf where the principal diurnal constituents are much larger than the semidiurnals, a response at the Mf frequency is readily observed (Marsh et al., 2013).**

*General: I think you need to be clear that Talutis and Carlson ice streams flow into what you only call "Carlson Inlet".*

**Carlson Ice Stream is not an approved name in any national gazetteers. We follow the convention of previous authors and refer to the stagnant glacier between Talutis and Rutford ice streams as Carlson inlet (i.e. the King (2011) reference in our paper, particularly Fig 1, and also 'Vaughan et al. (2008) Flow-switching and water piracy between Rutford Ice Stream and Carlson Inlet, West Antarctica.**

**Journal of Glaciology, 54 (184) 41-48'.**

*5.1-5.2: This seems like a good place to explain, or reiterate, that Msf does have the same frequency as the M2/S2-based spring-neap cycle, since it hints at why the nonlinearity that appears as Msf lateral motion might arise.*

**We now state explicitly in this sentence that the Msf has the same frequency as the spring-neap cycle.**

*7.11-7.23: (a) This would be clearer if it was organized in frequency: M2, K1, Msf, Ssa. Maybe a short paragraph for each. (b) Then, organize Fig. 5 to the write-up, and cite the specific panel we're meant to look at for each.*

**Done**

*7.26: "As the tide rotates". Which is "the tide" here? Msf, or general energy flux of the principal tides, mainly M2 and S2? But maybe you mean "Although" rather than "As" ? I'm trying to understand how the tidal rotation relates to the lead/lag between the spring-neap modulation and Msf phase, without invoking physics that you haven't explained.*

**This sentence has been rewritten to better convey what we mean. We choose to show the vertical tide downstream of FIS, rather than any other point, because here the vertical spring-neap cycle will lead in phase compared to all other ice streams (because the principle semidiurnal tides which interact to produce the spring-neap cycle rotate clockwise around the FRIS).**

*11.1-11.6: I'd disagree with the idea that interpolated fields "better show the importance of". Figure 5a is the honest answer. Maybe if you just used a wider color range for Fig. 5 ("jet" rather than new Matlab default), you could easily discuss everything from Fig. 5a.*

**We still feel that panel a works well to give the reader a clearer picture of how Msf amplitude varies across the entire shelf in a way that is not possible through**

**Fig 5. The points where we have data are clearly indicated in the figure to avoid misleading the reader.**

*11.7-11.9: As for the previous comment, the dot plots of Fig. 5 are more honest than the interpolated map, so I'd rather see the dot plot for MSF velocity (as a fraction of mean speed).*

**In this regard we agree with the reviewer, and have changed panel b from an interpolation to only show the point values for msf velocity.**

*11.7-11.16: I didn't understand this entire paragraph. (a) If the record is long enough to resolve Msf, then the detided signal has Msf removed, so the mean flow \*based on the detided residual\* should be unbiased by non-integer numbers of Msf in the record interval. The problem only arises if you do a standard mean of the non-detided record. (b) Why would the problem be worse in slower-moving parts of the ice sheet, especially if there is a reasonably linear relationship between Msf amplitude and mean speed (Figure 9)? (c) I really don't find Fig. 8b convincing, as the structure seems to be dominated by regions with poor data constraints.*

**(a) The reviewer is correct with regards to the Msf signal although longer periods such as the Ssa would have a very slight affect. The original statement now seems somewhat redundant and potentially confusing and so we have removed it. (b) This sentence has been removed as it is no longer relevant to the new figure. (c) Fig 8b has been changed as suggested**

*11.17-12.3: I think we need a bit more information here about \*why\* fortnightly variation \*always\* leads to increased mean speed regardless of mechanism. The three cited papers are all about ice \*streams\*; how does that map to, say, the Ronne ice front? Do you think that the Msf strain rates on the floating ice are increasing the mean flow, and increasing the mean strain?*

**The original statement is slightly flawed and so we have reworded this section.**

A nonlinear mechanism is needed to generate the Msf signal, i.e. horizontal ice flow ($u$) must have some nonlinear relationship to vertical ice-shelf displacements ($h$). This can be expressed most simply as $u \propto h^\alpha$ where $\alpha \neq 1$. Regardless of the mechanism being invoked, if $\alpha > 1$ the mean velocity will be increased, whereas a mechanism in which $\alpha < 1$ would reduce the mean velocity. All current theories that explain the Msf signal on both ice shelves and ice streams find that $\alpha > 1$.

*12.10-12.12: I wouldn't say that "the Ssa tides \*leads to\* a maximum in S2: they are part of the same component of astronomical forcing, and the description only makes sense if you are thinking S2 as being some modulated addition of S2 and K2.*

**Changed this to: The Ssa frequency is a long period constituent that results from the change in declination of the sun, which modulates the strength of the principal solar tidal constituent (S2).**

*12.19-12.20: My guess is that the statement that "FRIS is . . . the best observed . . . system" only really applies to the size of the GPS data base. For different aspects of ice shelves, more complete knowledge exists. It is probably good enough to just say that "The extensive data set of GPS records, and the large tidal ranges, . . . make the FRIS an excellent place to . . ."*

**Changed this sentence as suggested**

*12.28-12.29: "has been shown to lead to a higher . . ." Not true \*of this paper\*, and maybe not true in general for the ice shelf if the only basis is glaciological models for ice streams?*

**Reworded this sentence, firstly to remove the possible interpretation that we show this in our paper, and secondly to remove the implication that a nonlinear mechanism will \*always\* lead to higher mean ice velocity.**

MINOR COMMENTS

*1.3: (a) "motion" (of the ice shelf) in what direction? Clearly in the vertical, but less in the horizontal.*

**Specified that we refer to horizontal motion**

*(b) comma after 'shelf'*

**Done**

*1.10-1.13: Reorganize and/or subdivide the sentence.*

**We think the sentence works well as-is to convey the logical step-by-step arguments that lead to the final point of the abstract and is not overly long.**

*1.20: cite for the 'up to 9 m' statement. I think this comes from interpreted tilt rather than GPS.*

**We now can't track down a reference for the 9m range and so we have revised this statement to 7m and added a reference.**

*4.15-4.18: This paragraph should be merged with lines 4.3-4.7, to put the observations stuff together before summarizing the hypotheses for nonlinear effects.*

**Done**

*4.28: "overwinter" seems like an adjective. So, "continuous overwinter data . . ." or "data collection over the winter"*

**Changed to "over the winter"**

*4.33: Don't need "It is important to emphasise that"*

**Removed**

*5.7 and 5.11: Why 'FR-' sites?*

**Added an explanation that this refers to all sites whose names begin FR**

*5.12: Less colloquial way to say "these steps were skipped"*

**Changed to '. . .these corrections were not done. . .'**

*6.1: When you say that the Rayleigh criterion (lower case c) "was used", do you mean that it's a characteristic of the software, or \*you\* made the decision to use it?*

**Clarified that the Utide software makes the decision itself using this criterion**

*7.30: No need to hyphenate "time scales"*

**Removed hyphen**

*8.2-8.3: Sentence starting "The various" is just figure caption material.*

**Removed this sentence**

*8.4: capitalization of "data analysis" ?*

**Removed Capitalisation of data**

*10.6: Bracketing of citation "(Minchew et al., 2016)"*

**Fixed**

*11.9: formatting of 'Msf'*

**Fixed**

*12.16-12.17: This interesting issue (aliasing of InSAR-derived velocities) seems like it should have been mentioned in the Introduction.*

**This is mentioned in the introduction (4.6-4.7 of original submission)**

FIGURES

*F.1: (a) Add ice front to plot. Correct citation for CATS2008a is "an updated version of the inverse tide model described by Padman et al. (2002)"*

**The position of the ice front has been added in magenta, and the citation corrected.**

*F.2: (a) Caption should explain ice stream names, including that box-d is Talutis and Carlson ice streams. (b) "smoothed with a low-pass boxcar filter" should probably explain its characteristics (X km) and \*brief\* reason why.*

**Done**

*F.3: Part of the 'TIS' box is actually also "Carlson Ice Stream", isn't it?*

**Added Carlson Inlet to the box label.**

*F.6: For reasons I can't explain, I expected grounded ice to be on the \*left\* side of these plots. You could reverse them for me. Or, probably more sensible, add a vertical dashed line on each panel for the GL, then at the top of the plot mark "Ice Stream" and "Ice Shelf".*

**We have implemented both of these ideas as they both help make the figure clearer.**

*F.8: See Major Comments. I don't think this interpolated plot works well; the dots like Fig. 5 are better, especially for panel (b)*

**Panel b changed as suggested, for panel a see reply in major comments.**

*F.9: This might be more information-rich if you used different symbols for ice shelf and ice streams*

**This works well and we have also included separate $R^2$ values for ice shelf and ice stream sites.**